# The Study of Network Community Capacity to be a Subject: Digital Discursive Footprints

**DOI:** 10.3390/bs9120119

**Published:** 2019-11-21

**Authors:** Anatoly N. Voronin, Taisiya A. Grebenschikova, Tina A. Kubrak, Timofey A. Nestik, Natalya D. Pavlova

**Affiliations:** 1Laboratory of Speech Psychology and Psycholinguistics, Institute of Psychology, Russian Academy of Sciences; 13, Yaroslavskaya, Moscow 129366, Russia; gretiya@mail.ru (T.A.G.); kubrak.tina@gmail.com (T.A.K.); pavlova_natalya@mail.ru (N.D.P.); 2Laboratory of Social and Economic Psychology, Institute of Psychology, Russian Academy of Sciences; 13, Yaroslavskaya, Moscow 129366, Russia; nestik@gmail.com

**Keywords:** discourse, digital footprints, group reflexivity, network community, subjectness

## Abstract

The article is devoted to the assessment of the network community as a collective subject, as a group of interconnected and interdependent persons performing joint activities. According to the main research hypothesis, various forms of group subjectness, which determine its readiness for joint activities, are manifested in the discourse of the network community. Discourse constitutes a network community, mediates the interaction of its participants, represents ideas about the world, values, relationships, attitudes, sets patterns of behavior. A procedure is proposed for identifying discernible traces of the subjectness of a network community at various levels (lexical, semantic, content-analytical scales, etc.). The subjective structure of the network community is described based on experts’ implicit representations. The revealed components of the subjectness of network communities are compared with the characteristics of the subjectness of offline social groups. It is shown that the structure of the subjectness of network communities for some components is similar to the structure of the characteristics of the subjectness of offline social groups: the discourse of the network community represents a discussion of joint activities, group norms, and values, problems of civic identity. The specificity of network communities’ subjectness is revealed, which is manifested in the positive support of communication within the community, the identification and support of distinction between “us” and “them”. Two models of the relationship between discursive features and the construct “subjectness” are compared: additive-cumulative and additive. The equivalence of models is established based on the discriminativeness and the level of consistency with expert evaluation by external criteria.

## 1. Introduction

There are several approaches to studying the subjectness of communities, in each of which the following key notions of the approach as a whole stand out: the “aggregate subject” [1,2], “group subject” [3,4], “subject of joint activity” (A. V. Brushlinsky, V. V. Rubtsov, etc.), “collective subject” [4,5], “polysubject” [6,7], and others. The phenomenon of subjectness at the group level is revealed to the fullest extent possible through an analysis of a collective subject and its attributes such as joint activity, interconnectedness, interdependence, reflection [8]. The properties of a collective subject, which may have different levels of development, make it possible to reveal the mechanisms of the formation and functioning of an online community as a group of people interacting in the discursive space of the Internet, united by special connections and relationships and capable of manifesting joint forms of activity and self-reflection. A fundamental characteristic of any community is communication, and the discursive paradigm of research involving the study of real communicative practice in various situations and socio-cultural contexts [9,10,11,12,13,14] seems to be the most suitable in this regard.

A discourse of online communities is characterized by permanent publicity coupled with the intention to make a profitable self-presentation, while anonymous participation in group communication generates increased verbal aggression [15,16]. The material of news sites shows that the involvement of real people in online communication has a stronger influence on comments than the participation of non-personalized news service representatives under a site logo: the level of politeness and the desire for objectivity in comments increase [17]. It is emphasized that forums, where participants follow the norms of cooperative polite communication, are more meaningful. Under these conditions, the growth of knowledge, convergence of opposing views, reduction of the gap between attitudes, and behavior is demonstrated [18,19]. It is shown that some topics “attract” comments of one or another quality: health (healthcare) and crime-related problems raise specific questions considerably more often that are aimed at finding information/rating opinions, and more relevant and polite comments arise around education. The same economic topics cause a greater increase in unwarranted comments and ratings [17].

It is suggested that Internet communication manifests common social-psychological effects previously emphasized during direct intra-group interaction [20]: group polarization, when participants of Internet communication are eager to seek confirmation of their views [21], conformity to a leader that supports the standards of conduct for his or her group [22], the SIDE-phenomenon, which is associated with the name S. Moskovisi (social identity deindividution effects), according to which behavior is caused by standards that match the identity updated in this context [23,24]. It was shown that Internet content reproduces existing ethical standards and rules of its creators, reflects the traditional attitudes and preferences of this society [25]. Some specific phenomena receive coverage: flaming is a deliberate violation of the standards of communication on the Internet with the purpose of causing a negative reaction [26], cyberbullying is harassment common among adolescents [27], and others. Attempts are being made to describe and develop conditions for the implementation of standards of cooperative communication [15,28,29]. Compared with anonymous unregulated communication, the presence of conditions such as mandatory non-anonymity as well as pre-moderation of messages was shown to contribute to the implementation of standards of courtesy and mutual respect [15,16].

The study of online discourse is associated with the study of real communicative practices in different conditions and social interactions. Within the framework of discourse analysis, practices used to structure social and individual representations of speakers [30,31,32] are studied, influence is asserted, and power realized speakers [33,34,35,36], dialogical interaction of interlocutors and communication with the audience is organized [37,38,39].

Advancement in this direction involves developing an approach to the empirical study of online communities that combines social-psychological and psycholinguistic methods. Since the mechanisms of the formation and functioning of an online community as a group of people interacting in the discursive Internet space are revealed to the fullest extent possible through an analysis of the phenomenon of collective subjectness [8], the task was to identify the discursive characteristics of communities relevant to various forms of subjectness of a group: interconnectedness and interdependence of individuals in a group, the ability of the group to manifest various forms of joint activity, group self-reflection, i.e. group reflexivity [40,41]. Since the social-psychological status of an online community is not established, then the components of subjectness mentioned can only be regarded as guidelines.

According to the main hypothesis of the research, various forms of group subjectness manifest themselves in the discourse of an online community, constituting the community, mediating the interaction of its members, representing the world views, values, relationships, attitudes, and defining behavioral patterns. It is suggested to identify and verify the features of the discourse of online communities relevant to the various forms of subjectness by identifying individual discursive features of various levels (lexical, semantic, content-analytical scales, etc.) as digital discursive footprints of subjectness. To determine the structure of the subjectness of an online community according to discursive features and to compare different models of the interconnection between discursive features and the construct “subjectness” was the goal of the study.

## 2. Materials and Methods

### 2.1. Identifying Discursive Characteristics Based on a Psycholinguistic Analysis of the Corpus of Online Communities

The identification was done by 4 experts in psycholinguistics using a psycholinguistic analysis of the corpus of 2 online communities: an opposition political forum (https://politota.d3.ru/sapozhnik-bez-sapog-upd-1682202/?sorting=rating), genre: political discussions of like-minded people (opposition), Ford car owners forum, (https://forum.auto.ru/mark/ford/1574251/), genre: request for advice, community opinion. A discussion and coordination of methods for identification and coding were carried out during 5 expert sessions conducted using the method of grounded theory [42,43].

### 2.2. Expert Assessment of Discursive Features According to Subjectness Scales

To determine the implicit structure of the subjectness of online communities that manifests itself in the discourse, a subjective scaling of identified discursive features used to assess the subjectness of various offline collective subjects was conducted according to subjectness scales [8]. The procedure involved 6 psychologists specializing in the psychology of subjectness. The data of expert evaluation protocols, after checking consistency and eliminating obviously outlying data, were averaged and subjected to cluster analysis.

### 2.3. Evaluation of the Subjectness of Online Communities Based on Discursive Features

The subjectness of 6 online communities was assessed: 1) the FB (Facebook) Blue Buckets group, a platform for discussing issues of equality and violation of citizens’ rights on Russian roads (https://www.facebook.com/groups/bluebuckets), 2) an opposition political forum (https://politota.d3.ru/sapozhnik-bez-sapog-upd-1682202/?sorting=rating), 3) a Ford car owners forum (https://forum.auto.ru/mark/ford/1574251), 4) a community with a leader on the FB page of journalist and film critic A. Dolin (https://www.facebook.com/adolin3/posts/10217168128451538), 5) group chat in Telegramm “Progressors”, dedicated to relations between people and personal growth (https: //t.me/progressors), 6) a group in VK (“BK_онтакте_”) “The Suffering Middle Ages”, a platform for discussing in a sarcastic way “pain, suffering, and humiliation in the Medieval world and modern Russia” (https://vk.com/souffrantmittelalter). Among different digital footprints of online communities, the texts are the most relevant traces for evaluation of the subjectness. The texts of online communities were marked by four expert psycholinguists by highlighting discursive features. Subjectness was calculated in accordance with two models for the relationship between the features and constructs: additive and cumulative additive. A comparison of the models was carried out based on their discriminatory power. The subjectness calculated based on discursive features was compared with the subjectness calculated based on the expert evaluation.

### 2.4. Statistical Methods

For statistical calculations, the PASW (Predictive Analytics SoftWare) Statistics 18 package was used. The consistency of experts’ opinions was evaluated using Cronbach’s alpha, the latent structure of subjectness was obtained using the Hierarchical Cluster Analysis, the consistency of expert assessment of subjectness and subjectness, assessed with discursive features, was evaluated using the Kendall concordance coefficient W, and the discriminatory power of the models was evaluated as Squared Euclidian distance between the levels of subjectness.

## 3. Results and Discussion

### 3.1. Discursive Features of the Online Communities Subjectness 

The psycholinguistic analysis of the discourse of online communities allowed us to single out 76 features of subjectness such as “imperative statements including the speaker’s scope”, “indications of the number of group members”, “priority and forbidden topics for discussion”, “evaluative comments about communication in the online community”, “the use of vocabulary with semantics of abstraction and generalization”, “calls to action”, “nominations of occasional choice”, “the declared acceptance of roles.” At the initial stage of the analysis, identifying discursive features of various levels (lexical, semantic, content-analytical scales, and others) arranging the “empirical substance of the discourse of subjectness” and differentiating it from text fragments that do not present subjectness. During five expert sessions, primary coding was carried out, substantive and theoretical codes were determined. The pre-allocated discursive features of the subjectness of online communities were corrected and grouped into 7 blocks: 1) the interconnectedness of the participants of an online community, 2) the membership of the online community and its unity, 3) group social ideas about the online community, 4) the opposition “we–the others”, 5) psychological readiness for joint activity, 6) the presence of a common goal, 7) the manifestation of civic consciousness of the participants of the online community. Each block represented a certain theoretical code that functionally and structurally revealed a certain aspect of the subjectness of online communities. Based on some features, the experts had disagreements and doubts about their acceptability for assessing subjectness, which led to a reduction in the initial list to 64.

### 3.2. The Subjectness Structure of Online Communities Based on the Analysis of their Discourse

The six experts involved in the study of collective subjects participated in the final session on determining the discursive features of subjectness to get acquainted with their content and manifestations in the texts of online communities. At the next stage, they evaluated each of the 64 features according to different scales characterizing the subjectness of various social groups grouped into three blocks: 1) characteristics of a large social group (interconnectedness, joint, activity, reflection of life, interaction with members of other communities, etc.), 2) procedural characteristics of the collective subject of activity (common historical past of the community, the presence of a common language, the manifestation of social initiative on the part of the community, the presence of a relatively stable system of ideas and opinions, etc.), 3) characteristics of polysubjectness (in the form of dichotomies: cohesion-disunity, openness-closeness, compatibility-incompatibility, conflict-conflict-free) [8]. The evaluation was carried out according to a 5-point Likert scale describing to what extent a discursive feature determines a specific characteristic of subjectness. Thus, 6 matrices were obtained, each of which contained 55 characteristics of subjectness (columns) and 64 discursive features (rows). To summarize the results of the scaling, an assessment was conducted of the consistency of the expert opinions. For each scale, the Cronbach alpha α concordance coefficient was calculated. If α for 4 or more experts turned out to be > 0.7, then the results were considered to be matched, but the data of the experts that worsen the consistency assessment were not taken into account in the future. Based on nine characteristics, it was not possible to obtain sufficient consistency, and they were excluded from the analysis. The matched data were averaged over the experts and subjected to the cluster analysis procedure (Hierarchical Cluster Analysis, squared Euclidean distance measuring method, between-groups linkage clustering method). Figure 1 shows the dendrogram and the selected variant involving a split into clusters, taking into account the maximum distance when combining objects into clusters and the clarity of interpretation in a thoughtful generalization of the features included in a cluster. In total, 10 clusters of Level 1 and 2 of the second level were highlighted.

A thoughtful interpretation of the discursive features included in the clusters allowed us to reveal the following subjectness structure of online communities (Figure 2).

In essence, this is a reconstruction of the structure of implicit ideas about the subjectness of online communities of experts in the field of social psychology studying the properties of collective subjects included in various forms of online activity and understanding the special features of discourse characteristic of mass media. The proposed components of subjectness differ significantly from the characteristics of the subjectness of real social groups. According to A. Zhuravlev [8] there are three most important characteristics of a social group that are necessary and, in fact, criterial in describing a collective subject: 1) interconnectedness and interdependence of individuals in a group, 2) the ability of the group to manifest various forms of joint activity, i.e., to speak out, to be one whole in relation to other social objects or in relation to itself, 3) the ability of the group for self-reflection.

“Interconnectedness and interdependence” as such are not represented in the subjectness structure of the online community. This is understandable - the voluntary participation in an online community and the technical possibilities of social networks predetermine the interconnectedness and interdependence of networks. However, they do not guarantee the solidarity of a social group, which predetermines the social mechanisms of “protecting” a community from “outsiders”. In our case, by generalizing their discursive features, four such mechanisms were identified. Language and conceptual identification involve identifying one’s own by language features: the commonness of language and metaphors used, by “password words, by slang and special terms specific to this community. The identification of one’s own suggests a designation of others from which to distance oneself. The exclusion of others implies their labeling, discredit, condemnation, offense. The selection of others contributes to relaying the image of the online community. If this mechanism turns out to be ineffective, then community members enter into a dispute with the “outsiders”, impose bans on their activity, show aggression towards them. This mechanism was referred to as “protection of the online community”.

The discourse on joint activities is entirely comparable with such a characteristic of real collective subjects as the ability of a group to manifest various forms of joint activity. Also in the online community, group standards and values have been preserved as one of the leading characteristics of collective subjects - the group’s ability to self-reflect. Positive communication support is the most vividly presented component of online-community subjectness, most likely due to the technical features of social networks, initially focused on positive communication - orienting members of online communities to acquiring more and more friends, comments in the form of positive emoji, approval of posts with likes. Civic identity, a characteristic that is fully applicable to large social groups, is also entirely appropriate in the discourse of an online community since its size clearly exceeds the size of a small and medium-sized social group. Thus, the structure of the subjectness of online communities, evaluated using discursive features, has five basic components: 1) discussion of joint activities (group reflexivity), 2 – positive communication support, 3) reference to group standards and values, 4) supporting distinction between community members and outsiders, 5) construing and defending of civic identity.

### 3.3. Comparison of the Subjectness of Online Communities Based on Discursive Features

The texts of 6 online communities (FB Blue Buckets, Political Community, Ford Car Owners Forum, Leader Community, Progressors Group Chat, Medieval Suffering VK), including several posts in each and all comments on them were tagged by four expert psycho-linguists based on discursive features of subjectness. The subjectness of the communities was calculated based on two models: additive and cumulative additive. In accordance with the additive model (AM), the features detected were summed up in accordance with the subjectness structure, and only the presence or absence of a feature in the post or comments was assessed. In accordance with the additive-cumulative model (AKM), all features detected in the community text were summed up in accordance with the subjectness structure. The figures show histograms of averaged values of the subjectness of the online communities being assessed based on the five main subjectness features: 1) Discussion of joint activities, 2) Positive communication support, 3) Group standards and values, 4) One’s own–outsiders, 5) Civic identity (Figure 3, Figure 4). Due to differences in the number of posts and comments, the data obtained for these indicators were averaged. The general subjectness calculated according to the different models is presented in Figure 5. For the correctness of comparisons of the expressiveness of the general subjectness for indicators for different communities and models, a z-conversion of the data was carried out with a shift of negative values to 0.

An assessment was made of the consistency of the two models among themselves and with the data from the expert assessment of subjectness obtained by the group of social psychologists participating in the study at the previous stage. The data are presented in Table 1.

To assess the consistency between the subjectness assessed using the expert assessment and the subjectness obtained using discursive features, the Kendall concordance coefficient W was used. For the additive model, it was W = 0.971, for the additive-cumulative model, it was W = 0.914, which indicates an extremely high degree of consistency of both models with the expert assessment. The consistency of the models among themselves was W = 0.971.

The discriminatory power of the models was assessed through the average distance (squared Euclidean distance) between the levels of subjectness of the 6 online communities in the space of the five subjectness components. For the AKM model, the average distance =0.974, for the AM model = 0.915. In general, it can be said that the determination of subjectness using both models is entirely equivalent.

The model equivalence allows a certain way to relate to the subjectness of online communities and the organization of communication in them. Literally, following the description of the models, it can be concluded that the level of subjectness is most likely affected by a variety of means of its manifestation, not the number of repetitions of the same ways of initiating activity and subjectness. It is important to support communication in a variety of ways, not to literally insist on one’s own, it is important to call for action in different ways, not to repeatedly ask for material assistance, for example. To increase subjectness, it is important to support one’s own by all possible means and to “remove others” for various reasons. In light of the equality of the models considered, a repetition of the same slogans reduces subjectness rather than motivating it. No less important is the size of posts and comments: the equivalence of the models predetermines an equal influence on the subjectness of an online community of both short comments and posts and long ones.

The consistency of the assessment of the subjectness of online communities based on its “external” manifestations (off and online actions, collective coping, the declaration of community values and the discussion of communication rules in it, etc.), according to the expert assessments and the subjectness defined by the discursive features, makes it possible to talk about the reliability of the approach described for studying subjectness and the validity of the subjectness structure selected. The components of the subjectness of an online community only partially coincide with the characteristics of the subjectness of real social groups and reveal a new content of communicative activity in online communities: permanent support of communication, labeling of “outsiders” and their removal, identification, and support of ingroup members. Note should be made of a specific “civic identity” in the online communities studied. Civic identity is manifested in the opposition of active members of these communities and the condemnation of formally existing standards in society: capturing and discussing violations committed by representatives of power structures in the area of road traffic (typical of the blue bucket community) and the difficult conditions of existence of “ordinary people” in modern Russia (community “Suffering Middle Ages”). The opposition of the political community turned out to be less pronounced than expected by the experts, apparently as a consequence of declaring their own opposition and shifting the focus of communication activity to organizing assistance to “victims”.

## 4. Conclusions

In the discourse of online communities, various forms of their subjectness manifest themselves. Marking text in accordance with the selected discursive features allows you to adequately assess the subjectness of online communities. The subjectness structure of online communities is partially (for some components) similar to the structure of the subjectness characteristics of offline social groups: the discourse represents group reflexivity, group standards and values, and problems of civic identity. The specificity of the subjectness of network communities is revealed, which is manifested in the positive support of communication within the community, the identification and support of the distinction between “us” and “them”. Two models of the relationship between discursive features and construct “subjectness” are compared: additive-cumulative and additive. The equivalence of the models for discriminatory power and the level of consistency with the expert assessment based on external criteria is established.

## Figures and Tables

**Figure 1 behavsci-09-00119-f001:**
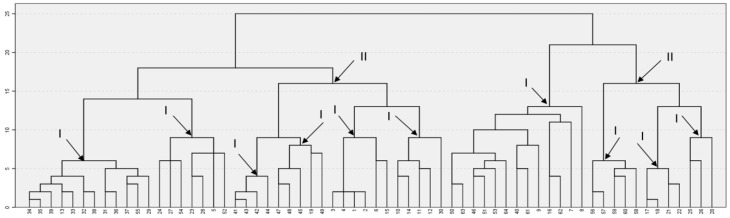
Dendrogram of discursive footprints and selected clusters. (**I**) the first level of clustering, (**II**) the second level of clustering.

**Figure 2 behavsci-09-00119-f002:**
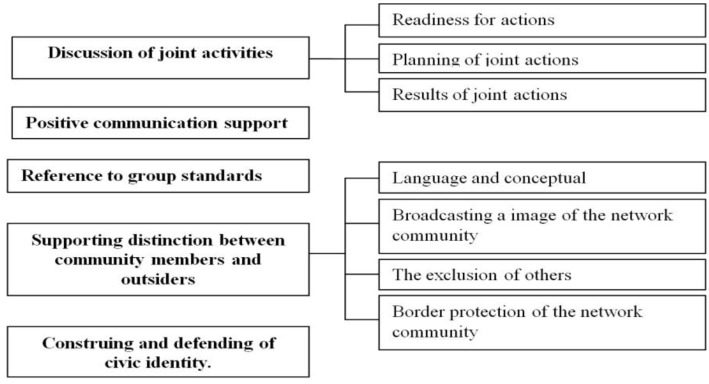
Сomponents of the subjectness of network communities.

**Figure 3 behavsci-09-00119-f003:**
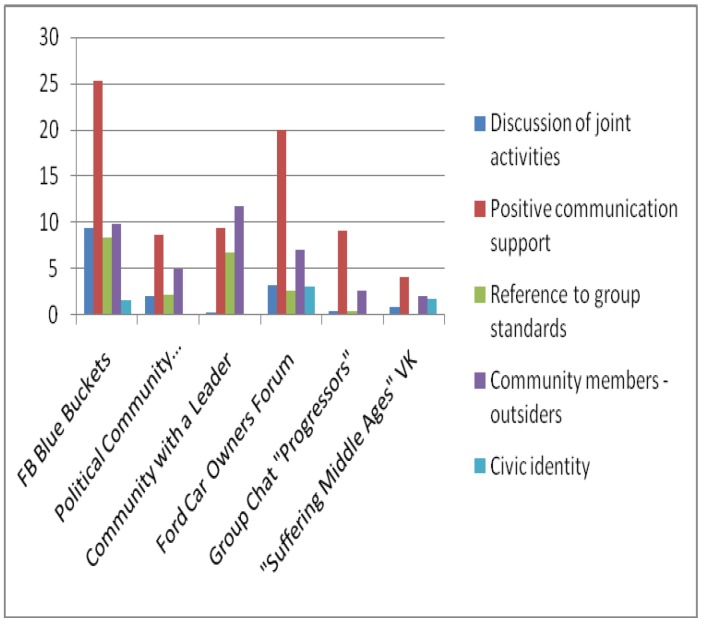
Subjectness additive-cumulative model (AKM).

**Figure 4 behavsci-09-00119-f004:**
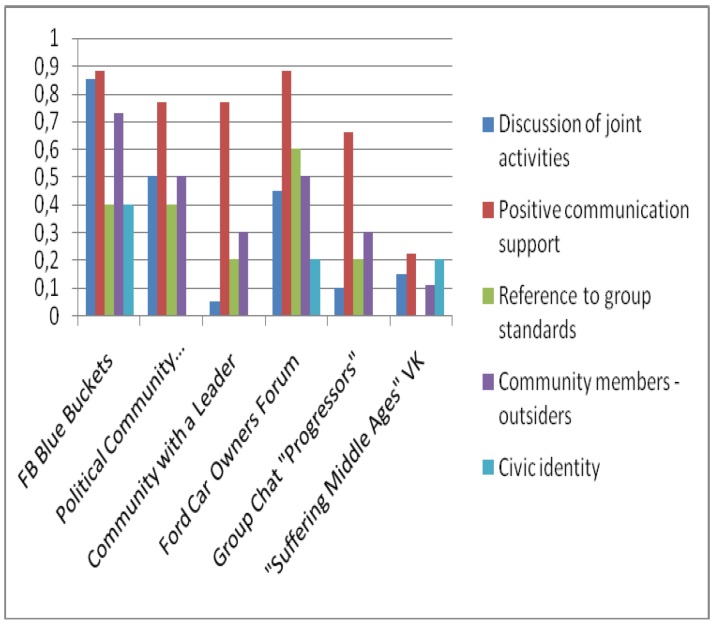
Community subjectness additive model (AM).

**Figure 5 behavsci-09-00119-f005:**
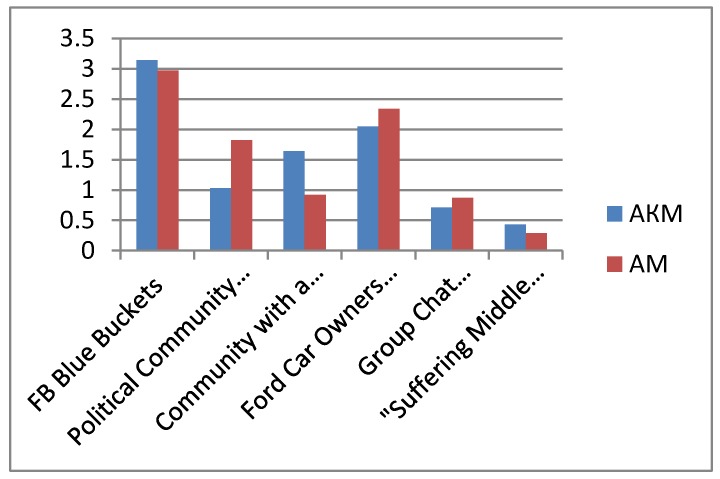
The general subjectness calculated based on the AKM and AM.

**Table 1 behavsci-09-00119-t001:** General subjectness of the network community (ranks).

Network Community	Model AKM	Model AM	Expert Evaluation
FB Blue Buckets	1	1	1
Political Community politota.d3	4	3	2
Community with a Leader	2	2	3
Ford Car Owners Forum	3	4	4
Group Chat “Progressors”	5	5	5
“Suffering Middle Ages” VK	6	6	6

## Data Availability

The datasets used and analyzed during the current study are available from the corresponding author on reasonable request.

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
