# Peer review of "The Study of Network Community Capacity to be a Subject: Digital Discursive Footprints"

_behavsci, 2019, doi:10.3390/bs9120119_

Round 1

Reviewer 1 Report

Interesting study. However, much needs to be corrected and explained better. Anyway, the authors should surely correct these:

Line 37:  “subject of joint activity” (A. V. Brushlinsky, V. V. Rubtsov, etc.)

and Line 38 (the way of ref)  [4, 5, et al.], “polysubject” [6, 7, et al.], and others

Line 46 … and similar ones through the whole text.

Line 115: (Zhuravlev, 2018)

Line 172: “matrices in which the characteristics of subjectness (55) are combined with discursive features (64).” – what is (55) and (64)?

Chapters 3.1. and 3.3. should be rewritten (and without hyperlinks)!

Line 137: All analyzes that have been carried out (and for what purposes), should be indicated here.

The Goal should be integrated into Introduction section.

The Results and Discussion sections should be apart.

The reference list should be updated.

Reviewer 2 Report

Although my expertise does not fully cover the aspects of this psycholinguistic research, I think the work is well done. The qualitative analysis that underlies this research is well done, supported by appropriate statistical methods and presented clearly.

It might have been useful to specify the field of expertise of the experts: linguists, psychologists, psycholinguists, etc., as well as some descriptive characteristics of each corpus analyzed.
The work is interesting and can be published in its state.

Best regards

Round 2

Reviewer 1 Report

I believe that the article is now more suitable for being published.

But, please, do update the reference list a little bit.

Author Response

Response to Reviewer 1 Comments

Point 1:

But, please, do update the reference list a little bit.

Response 1:

Thank you for the evaluation of our study, but we would like to clarify what the reviewer meant specifically when recommending updating the reference list.